# Evaluating the quality of evidence for gaming disorder: A summary of systematic reviews of associations between gaming disorder and depression or anxiety

**Michelle Colder Carras**[1¤]*, **Jing Shi**[2,3], **Gregory Hard**[4], **Ian J. Saldanha**[5]

1 Behavioral Sciences Institute, Radboud University, Nijmegen, Netherlands, 2 Institute for Mental Health Policy Research, Centre for Addiction and Mental Health, Toronto, Ontario, Canada, 3 School of Rehabilitation Science, Faculty of Health Sciences, McMaster University, Hamilton, Ontario, Canada, 4 MGH Institute of Health Professions, Mass General Brigham, Boston, Massachusetts, United States of America, 5 Center for Evidence Synthesis in Health, Department of Health Services, Policy, and Practice, and Department of Epidemiology, Brown University School of Public Health, Providence, Rhode Island, United States of America

¤ Current address: International Health, Johns Hopkins Bloomberg School of Public Health, Baltimore, Maryland, United States of America
* mcarras@jhu.edu

**Data Availability Statement:** All relevant data are within the manuscript and its Supporting Information files.

## Abstract

Gaming disorder has been described as an urgent public health problem and has garnered many systematic reviews of its associations with other health conditions. However, review methodology can contribute to bias in the conclusions, leading to research, policy, and patient care that are not truly evidence-based. This study followed a pre-registered protocol (PROSPERO 2018 CRD42018090651) with the objective of identifying reliable and method-ologically-rigorous systematic reviews that examine the associations between gaming disor-der and depression or anxiety in any population. We searched PubMed and PsycInfo for published systematic reviews and the gray literature for unpublished systematic reviews as of June 24, 2020. Reviews were classified as reliable according to several quality criteria, such as whether they conducted a risk of bias assessment of studies and whether they clearly described how outcomes from each study were selected. We assessed possible selective outcome reporting among the reviews. Seven reviews that included a total of 196 studies met inclusion criteria. The overall number of participants was not calculable because not all reviews reported these data. All reviews specified eligibility criteria for studies, but not for outcomes within studies. Only one review assessed risk of bias. Evidence of selective outcome reporting was found in all reviews—only one review incorporated any of the null findings from studies it included. Thus, none were classified as reliable according to pre-specified quality criteria. Systematic reviews related to gaming disorder do not meet meth-odological standards. As clinical and policy decisions are heavily reliant on reliable, accurate, and unbiased evidence synthesis; researchers, clinicians, and policymakers should consider the implications of selective outcome reporting. Limitations of the current summary include using counts of associations and restricting to systematic reviews published in English. Systematic reviewers should follow established guidelines for review

**Funding:** The author(s) received no specific funding for this work.

**Competing interests:** I have read the journal's policy and the authors of this manuscript have the following competing interests: Dr. Colder Carras has spoken to the World Health Organization about this topic at the request of the Entertainment Software Association but has received no funding, honorarium, fees, meals, lodging, donations or reimbursement. She has also spoken about this topic to other audiences: She received an honorarium and airfare to speak to the Johns Hopkins Department of Psychiatry about video games and mental health and has addressed this topic in other unpaid/unreimbursed conferences and lectures. Dr. Colder Carras has no past, current, or planned business or financial relationships with any other organization related to the video game industry. She has acted as a consultant with a nonprofit organization, Stack Up, that provides mental health and suicide prevention support through online video game play. She is currently a member of the American Association for Suicidology Technology and Innovation Committee and believes that video games and social media have the potential to be useful platforms for delivering public health interventions. She sometimes plays video games herself. The other authors declare no conflict of interest.

conduct and transparent reporting to ensure evidence about technology use disorders is reliable.

## Introduction

Gaming disorder or Internet gaming disorder (IGD) is a disorder related to excessive video, computer, or online game play that results in psychological distress and/or functional impairment [1,2]. Internet gaming disorder was included as a condition for further research in the 5th edition of the Diagnostic and Statistical Manual (DSM-5) and the diagnosis of gaming disorder has been added to the 11th edition of the World Health Organization (WHO) International Classification of Diseases (ICD-11) [1,2]. Gaming disorder includes symptoms related to substance use disorder, such as loss of control (that continues despite negative consequences), functional impairment, distress, and/or interference with daily activities. The disorder is distinguished from other related disorders, such as technology overuse, Internet addiction, and social networking addiction [3]. Recent commentaries have described gaming disorder (which we will define here broadly to include the diagnoses of IGD or gaming disorder, problematic/pathological video gaming, and other concepts related to excessive video game play) as a clinical and public health problem in urgent need of advancements in treatment development [4,5].

Delineation and measurement of a clear construct with no overlap with other related conditions, such as gambling, Internet use, and technology use, are crucial to this field. Many recent commentaries on the need for a diagnosis of gaming disorder use terms like "Internet addiction or Gaming disorder" [6], "Internet-related disorders including gaming disorder" [4], and "Internet addiction including gaming addiction" [5], pointing to the persistent overlap in measurement of these problematic behaviors. From a public health perspective, many forms of Internet use—not just gaming—continue to be recognized as potentially problematic, as evidenced by a recently-funded international research collaborative on problematic Internet use [7].

Systematic reviews are research activities that follow established, rigorous methods to summarize all relevant evidence on specific research questions that are vital for decision-making by clinicians, patients, policy-makers, and other stakeholders. The methods include framing the research question, searching for the evidence, screening studies for eligibility, assessing risk of bias and extracting data from included studies, conducting qualitative and, where merited, quantitative syntheses, and reporting the findings. Recent decades have witnessed a surge in the number of systematic reviews conducted [8]. Multiple standards have been developed for the conduct and reporting of systematic reviews [9]. However, research has shown that reviews in some fields provide low-quality evidence, are unreliable, and can be sources of bias themselves [8,10,11]. Bias can sometimes be introduced due to methods used in the systematic review ('meta-bias') [12].

One source of meta-bias can potentially occur when a given study included in a review reports results for a given relevant outcome in multiple ways, and the reviewer must make a choice among these to determine which result(s) to extract for the review [13,14]. In this situation, choice of the result based on the largest (or smallest) magnitude of treatment effect, on statistical significance, and/or on the result that supports the reviewer's conscious or subconscious preconceptions can be problematic and lead to bias. Such bias can be preempted by completely prespecifying the five elements of an outcome (Fig 1) [10,15]. However, complete

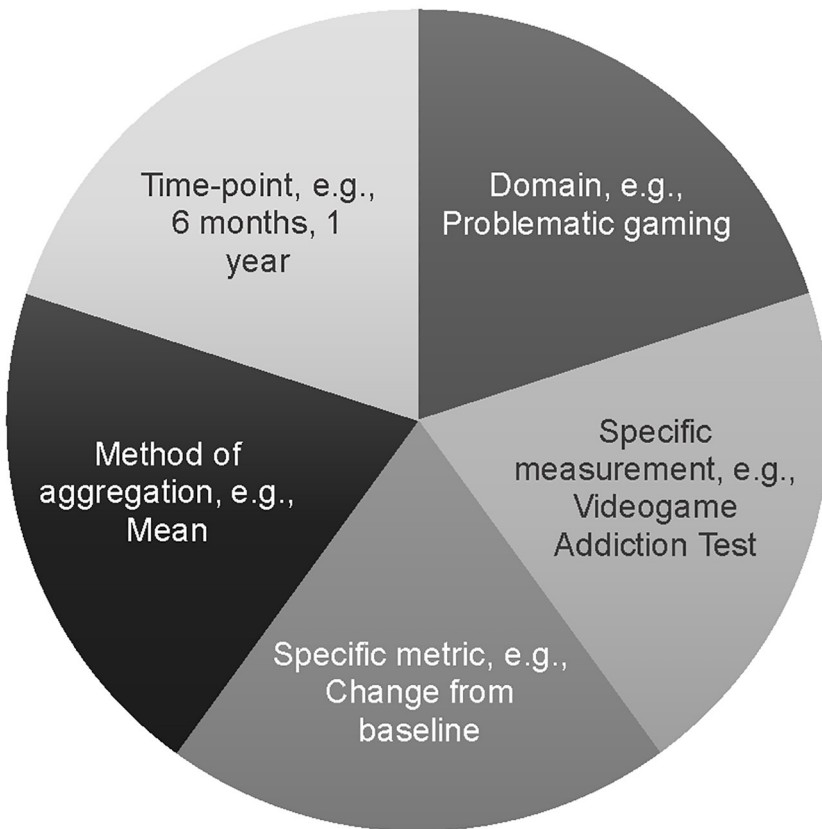

**Fig 1. Defining outcomes for a systematic review or meta-analysis.** Elements of outcome domains required for complete outcome specification in health research. Figure adapted from [15]; see also the PRISMA-P [16] statement or description of PICOS [15].

prespecification is not always possible and/or may be considered too restrictive. Moreover, choosing specific results from multiple reported analyses from multiple data sources for a given study is a multi-dimensional problem. In one study of meta-analytic methods, an examination of outcomes reported in 14 clinical trials revealed that, depending on which outcomes from the trials were chosen by the reviewers, over 34 trillion meta-analyses were possible [13].

Now that gaming disorder has been recognized as a disorder by the WHO, ensuring systematic and accurate measurement of gaming disorder in studies *and* accurate reporting of exposures, outcomes, and conclusions in reviews are vital for ongoing decision-making regarding diagnosis, treatment, and public health interventions. Given the established association between gaming disorder and two common mental health outcomes—depression and anxiety—we limited the scope of our study to systematic reviews that included data about these outcomes. This allowed us to explore the issue of selective outcome reporting in reviews.

In this summary of systematic reviews, we assess the reliability of current reviews that have examined the association between gaming disorder and depression or between gaming disorder and anxiety in any population. We aimed to answer the following research questions to inform directions for future research and policymaking:

1. Do systematic reviews of the associations between gaming disorder and depression and between gaming disorder and anxiety meet reliability standards for systematic reviews?

2. Do systematic reviews of the associations between gaming disorder and depression and between gaming disorder and anxiety distinguish between gaming disorder and other constructs, such as Internet addiction?

3. Do systematic reviews of the associations between gaming disorder and depression and between gaming disorder and anxiety report outcomes selectively?

4. What are the associations between gaming disorder and depression and between gaming disorder and anxiety reported in reliable systematic reviews?

## Methods

This study is a summary of systematic reviews of the associations between gaming disorder and depression and between gaming disorder and anxiety in any population. The review methods, including the research question, search strategy, inclusion/exclusion criteria, and risk of bias assessment, were developed *a priori* and described in the registered protocol (PROSPERO 2018 CRD42018090651); these are also available in S1 Protocol. All data, the protocol, a list of articles excluded at the full-text screening stage with reasons for exclusion, and other supporting documentation are available on our Open Science Framework website (*see Project on OSF website*) and in Supporting Information files. In this paper, we discuss two groups of research studies: the systematic reviews (henceforth called 'reviews') and the primary studies included in those reviews (henceforth called 'studies').

We examined reviews that included studies of the associations between the exposure of gaming disorder (as defined by the review authors) and the outcomes of depression or anxiety. We restricted to reviews published in English by June 24, 2020. We excluded reviews that:

- Were narrative reviews, overviews of reviews, commentaries, and other non-systematic reviews of studies;

- Only examined Internet addiction or other technological addiction; or

- Did not report results for the associations between gaming disorder and depression or anxiety separately (e.g., we excluded reviews that only reported pooled outcomes for "mental health").

Fig 2 illustrates how we defined the domains of depression and anxiety in our study. For the outcome of depression, we restricted to scales, subscales, diagnosis, or clinical interviews for depression or more severe single symptoms related to depression, such as suicidal ideation, but excluded measurements of nonspecific symptoms, such as low energy, sleep problems, sadness, or withdrawal from social activities. For the outcome of anxiety, we included scales, subscales, diagnosis, or clinical interviews for anxiety, social anxiety, and social phobia, but excluded measurements that combined anxiety with other constructs (e.g., anxiety/depression).

### Search strategy and screening process

We conducted electronic searches of PubMed and PsycInfo for published reviews and meta-analyses (searches were current as of June 24, 2020). Searches combined terms related to gaming disorder and terms related to depression or anxiety (S1 Search Strategies). In addition, we reviewed all years of the *Journal of Behavioral Addictions*, including its supplements, and all proceedings of the International Conference on Behavioral Addictions.

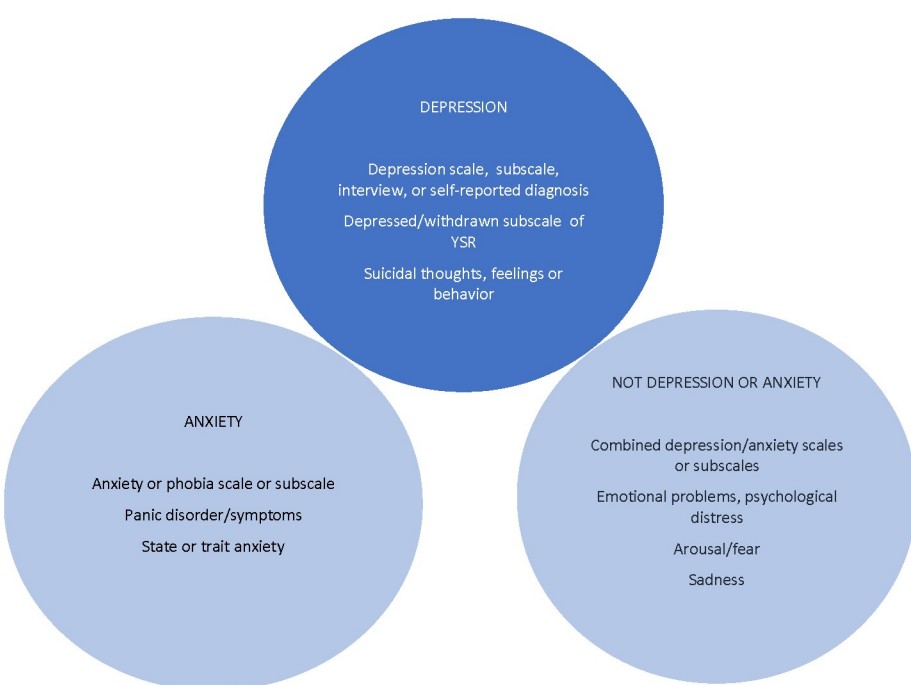

**Fig 2. Domains used to define depression and anxiety as constructs for analysis.** YSR = Youth Self Report scale.

## Assessment of reliability of reviews

We adapted the definition of "reliability" of systematic reviews developed by Cochrane Eyes and Vision [17–21]. This definition, in turn, was informed by items identified from the Critical Appraisal Skills Programme (CASP), A Measurement Tool to Assess systematic Reviews (AMSTAR), and the Preferred Reporting Items for Systematic Reviews and Meta-Analyses (PRISMA) tools [9,22,23]. According to this definition, a review is reliable when its authors did each of the following:

(1). Defined eligibility criteria for including studies;

(2). Conducted a comprehensive literature search for studies (i.e., searched at least one relevant electronic database, such as PubMed and PsycInfo; used at least one other method of searching, such as searching the grey literature, searching for unpublished studies, and searching the reference lists of included articles; and were not limited to English language citations);

(3). Assessed risk of bias in individual included studies;

(4). Used appropriate methods for meta-analysis, when conducted (e.g., adequately accounting for any heterogeneity); and

(5). Presented conclusions that were supported by the evidence reported in the review.

Because we also examined each study included in the reviews, we added an additional criterion that review authors should have:

(6). Specified in the methods or protocol which outcomes from their eligible studies were included in the synthesis or synthesized *all* reported outcomes from each included study.

We classified a review as reliable only if *all six* of the criteria were met. Finally, we conducted a full assessment of the quality of the included reviews using A Measurement Tool to Assess systematic Reviews—version 2 (AMSTAR 2) [24]; the full results of this assessment are found in S1 Data Extraction.

## Assessment of other outcomes

Other outcomes included the proportion of all studies within a review that measured gaming disorder with a gaming disorder-specific instrument; the proportion of reviews that specified all elements of an outcome; and the specific review- and study-level associations between gaming disorder and depression and anxiety. All reported associations within the studies were extracted from the original study reports and characterized as present and positive, present and negative, present and null, unclear, or absent. The count and type (positive, null, negative, unclear, or absent) of results for each study were compared with the results reported for each study in the reviews. We also made several comparisons regarding overall conclusions about the associations between gaming disorder and depression and between gaming disorder and anxiety by comparing bivariable versus multivariable analyses, cross-sectional versus longitudinal analyses, and results from reviews classified as reliable versus results from all reviews.

## Data extraction

We developed and pilot tested a data extraction form using Microsoft Excel®, based on the form developed by Mayo-Wilson et al. [17]. We added questions relevant to reviews of epidemiological studies [25]. During the initial data extraction, we noticed discrepancies in how specific studies were reported in the reviews, resulting in potential selective outcome reporting at the review level. To ensure that we evaluated this potential source of bias, we expanded the scope of our preregistered protocol to include examining study-level outcomes and how they were reported in reviews.

Two investigators from among MCC, JS, and GH extracted data from each review, consulting the third investigator for resolution of discrepancies where needed. If a review did not have a summary of findings table that included the total number of studies mentioned in the results or in supplementary material, we extracted data for all studies mentioned in text or tables of the Results section. Data on depression and anxiety outcomes within each study of each review were extracted by one investigator. Extracted data for a 10% random sample of studies were validated by the second and third investigator.

Data extracted from the reviews included information on methods for specifying eligibility criteria and outcomes, specific measurements (e.g., scales) of depression and anxiety in included studies, analyses conducted, whether and how review authors assessed risk of bias in included studies, specific measurement (e.g., scales) of gaming disorder in included studies, and all items from the AMSTAR 2 tool.

We summarize below the three conditions that had to be met for a specific measurement or scale to be classified as asssesing gaming disorder (Fig 3):

- The specific measurement or scale asked questions about computer, video, online, or digital game use in general, rather than just a single game (e.g., World of Warcraft®).

- The specific measurement or scale asked questions about gaming or online gaming rather than Internet or computer use in general (e.g., did not use only an Internet addiction measure, such as the Young Internet Addiction Test or the Compulsive Internet Use Test). If a study mentioned adapting a scale for video games and gave an example of an adapted question, we classified that scale as measuring gaming disorder. Otherwise, we classified the

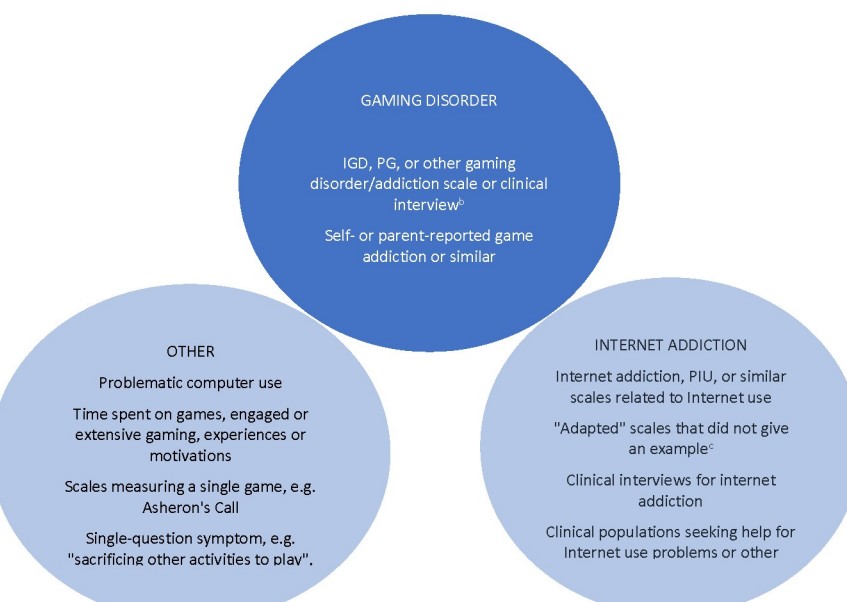

**Fig 3. Domains used to define gaming disorder as a construct for analysis.** IGD = Internet gaming disorder; PG = problematic gaming; PIU = problematic Internet use. (a) Sensitivity analysis: Clinical population of those seeking help for gaming-related problems but an Internet addiction scale was used. (b) Including those adapted from Internet addiction scales where an example question is given. (c) Where scales referenced appendices or other papers, these were also searched for example questions.

measurement according to the original scale from which it was adapted. We also conducted a sensitivity analysis to examine how our findings differed when other measurements (e.g., the Young Internet Addiction Test) were used with a clinical population diagnosed with gaming disorder. When the clinical population was unclear or was not diagnosed with gaming disorder and Internet addiction scales or other specific measurements/scales/interviews were used, we did not characterize this as gaming disorder (e.g., Young Internet Addiction Test in a clinical population of patients with gambling disorder).

- The specific measurement or scale asked questions about specific symptoms of gaming disorder rather than only experiences related to video game use in general, such as time spent playing games or the experience of time loss.

Data on depression and anxiety consisted of study scale, type of analysis, direction of association (positive, negative, or null), and how each review reported the outcome of the study (positive, negative, null, unclear, or absent).

## Quality assessment

See section above entitled 'Assessment of reliability of reviews'.

## Strategy for data synthesis and reporting

We narratively describe the characteristics of included reviews and their reliability. Because measurements of exposures and outcomes were heterogeneous, we present counts of positive or null/negative outcomes from studies and how they were reported in reviews [26]. Because consistency is one factor that supports strength of evidence, we compared tallies of qualitative associations from the multiple outcomes reported in studies. We described associations to be

'positive and consistent' at the study level if the count of statistically-significant positive associations was greater than the total number of negative or null associations. We described an association as 'null' if there were more null findings or negative associations than positive. We conducted a sensitivity analysis to examine the impact of measuring gaming disorder with a scale for Internet addiction in a clinical population of individuals with gaming disorder. All extracted data and derived variables are available in S1 Dataset.

The PRISMA checklist [9] for the current study is available in S1 PRISMA Checklist. This study was conducted using publicly-available information and therefore did not require Institutional Board (IRB) approval.

## Results

The searches yielded 842 records, of which, seven reviews were eligible for inclusion in this overview (Fig 4). The most frequent reasons for excluding articles (at the full-text screening stage) were that they were not a systematic review or did not specify methods (n = 35), did not report associations between gaming disorder and anxiety or depression (n = 23), and were not specific to gaming disorder (e.g., being about behavioral addictions in general) (n = 9).

### Review characteristics

The characteristics of the seven included reviews are reported in Table 1. They included a total of 196 unique studies. The number of included studies per review ranged from 24 to 63, with a mean of 46. Most studies (61.7%) were included in only one review each.

### Research question 1: Assessment of review reliability

We found that none of the seven included reviews fulfilled all six criteria for reliability. All reviews defined eligibility criteria and most reviews (six of seven) conducted comprehensive database searches (Table 1). No review defined outcomes using all five elements of completely-specified outcomes (i.e., domain, specific measurement, specific metric, method of aggregation, and time points). No reviews specified which outcomes of a study would be used in synthesis. One review specified that it would consider only study effect sizes from multivariable analyses, classifying full associations as ". . .a correlation was found for both genders after multivariable analyses" or partial associations as ". . .correlation was identified for only one gender" [28]. Other reviews did not specify how outcomes would be included, although some mentioned that "factors", "disorders", "comorbidity", "health-related outcomes", or "psychosocial features" "associated with" problematic gaming were "identified" [28], "ascertained" [29], or "extracted" [30,31].

Although all reviews acknowledged heterogeneity in measurement of problematic gaming, only one review assessed risk of bias systematically [30]. In this context, because five studies chose to conduct qualitative syntheses instead of quantitative syntheses (i.e., meta-analyses), we considered their results to have been combined appropriately. In one review, results were combined quantitatively despite a very high amount of statistical heterogeneity among studies (suggested by an $I^2$ value of 98%) [30]. Another review classified effect sizes as small, medium, or large and presented a table of counts of effect sizes for four mental health outcomes as a way to address heterogeneity in measurement [28]. Most reviews discussed limitations at the study, outcome, and review level, but two reviews did not discuss limitations systematically [27,29].

Assessment of AMSTAR 2 criteria showed that no study met all criteria, and some criteria were lacking in all studies. Full results can be found in S1 Data Extraction.

Because of the lack of clarity around how study outcomes were selected, the reporting of outcomes that was inconsistent with study findings (see Figs 5 and 6), the inclusion of studies

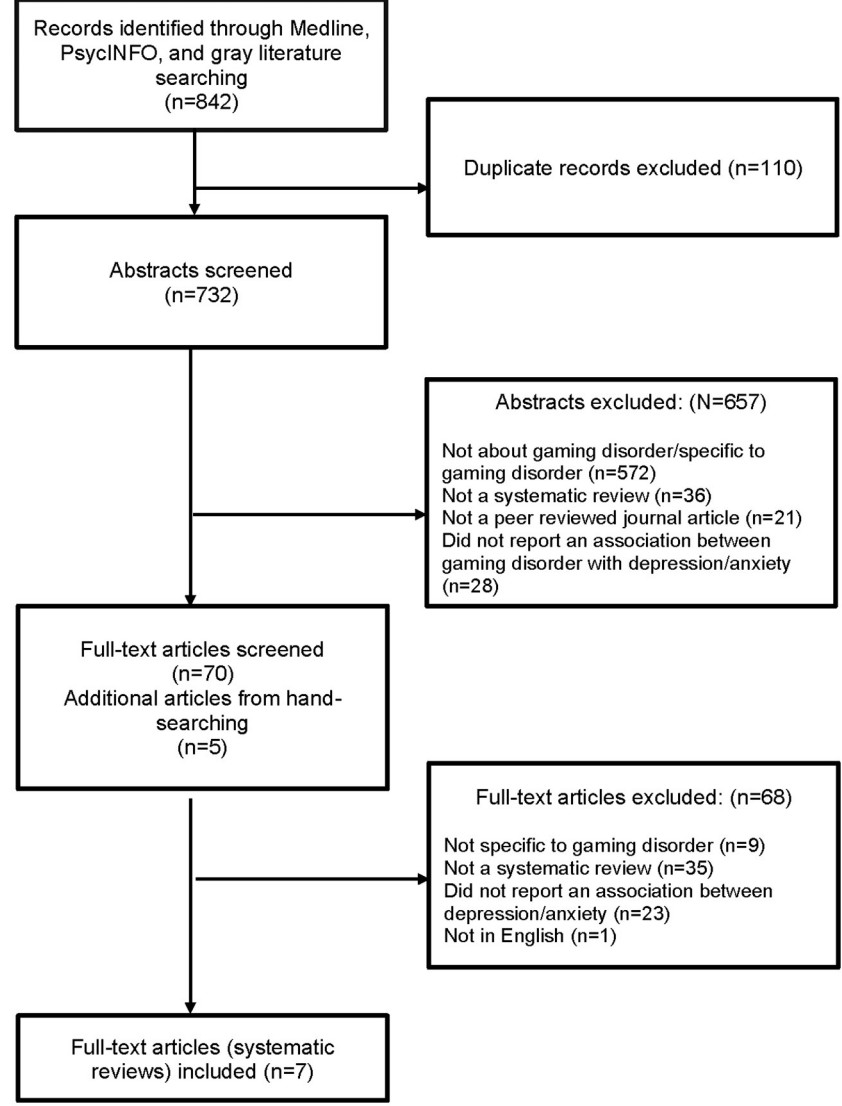

**Fig 4. PRISMA flow diagram.**

that did not measure gaming disorder, and the lack of systematic assessment of bias (except for one review [30]), we determined that review conclusions were not supported by the evidence from included studies. This is further explored in the following sections.

## Research question 2: Distinguishing between gaming disorder and other concepts

Based on our definition for measurement of gaming disorder (Fig 3), no review focused *only* on studies that measured gaming disorder. The percentage of studies within a review that measured gaming disorder ranged from 56.8% to 93.6%. On sensitivity analysis, where measurement of gaming disorder also included using an Internet addiction scale in a gaming disorder clinical population, the percent remained similar, ranging from 58.6% to 93.6%.

**Table 1. Review characteristics and reliability criteria.**

| First author, year | Number of included studies[a] | Total number of participants across all included studies[b] | Participant populations | Years of publication of included studies[c] | Number (%) studies measuring problematic gaming[d] | Reliability criteria | | | | | |
|---|---|---|---|---|---|---|---|---|---|---|---|
| | | | | | | (1) Defined eligibility criteria? | (2) Conducted a comprehensive search? | (3) Assessed of risk of bias? | (4) Used appropriate methods to combine results?[e] | (5) Conclusions about depression and anxiety supported by evidence? | (6) Specified which outcomes would be included in the synthesis? |
| Sugaya 2019 [27] | 51 | Unclear | Age 0–28 | Until 2018 | 25(56.8) | Yes | Yes | No | Yes | No | No |
| González-Bueso 2018 [28] | 24 | 53,889 | Any | 2011–2017 | 17 (70.8) | Yes | Yes | No | Unclear | No | No |
| Mihara 2017 [29] | 47 | 127,749 | Any | Until 2016 | 44 (93.6) | Yes | No | No | Yes | No | No |
| Männikkö 2017 [30] | 50 | 129,430 | >12.5 years | 2005–2016 | 41 (82.0) | Yes | Yes | Yes | No | No | No |
| King 2013 [31] | 63 | 58,415 | Any | 2000–2012 | 39 (61.9) | Yes | Yes | No | Yes | No | No |
| Kuss 2012 [32] | 58 | Unclear | Any | 2000–2010 | 33 (56.9) | Yes | No | No | Yes | No | No |
| Kuss 2012 [33] | 30 | 72,825 | Children | 2000–2011 | 20 (66.7) | Yes | Yes | No | Yes | No | No |

Notes:

(a) The number of included studies for a review is taken from the PRISMA flow diagram (where possible) or from reports in the text or tables of each review.

(b) The number of participants was taken directly from the text where possible or calculated from other information that was reported in the review.

(c) If no years were given, the end year was listed as one year prior to the year of publication.

(d) Proportion of studies measuring problematic gaming was assessed out of all studies mentioned in the review. This did not always match the number of studies that were said to be included in the review in the abstract, methods, or results.

(e) most reviews did not combine results quantitatively.

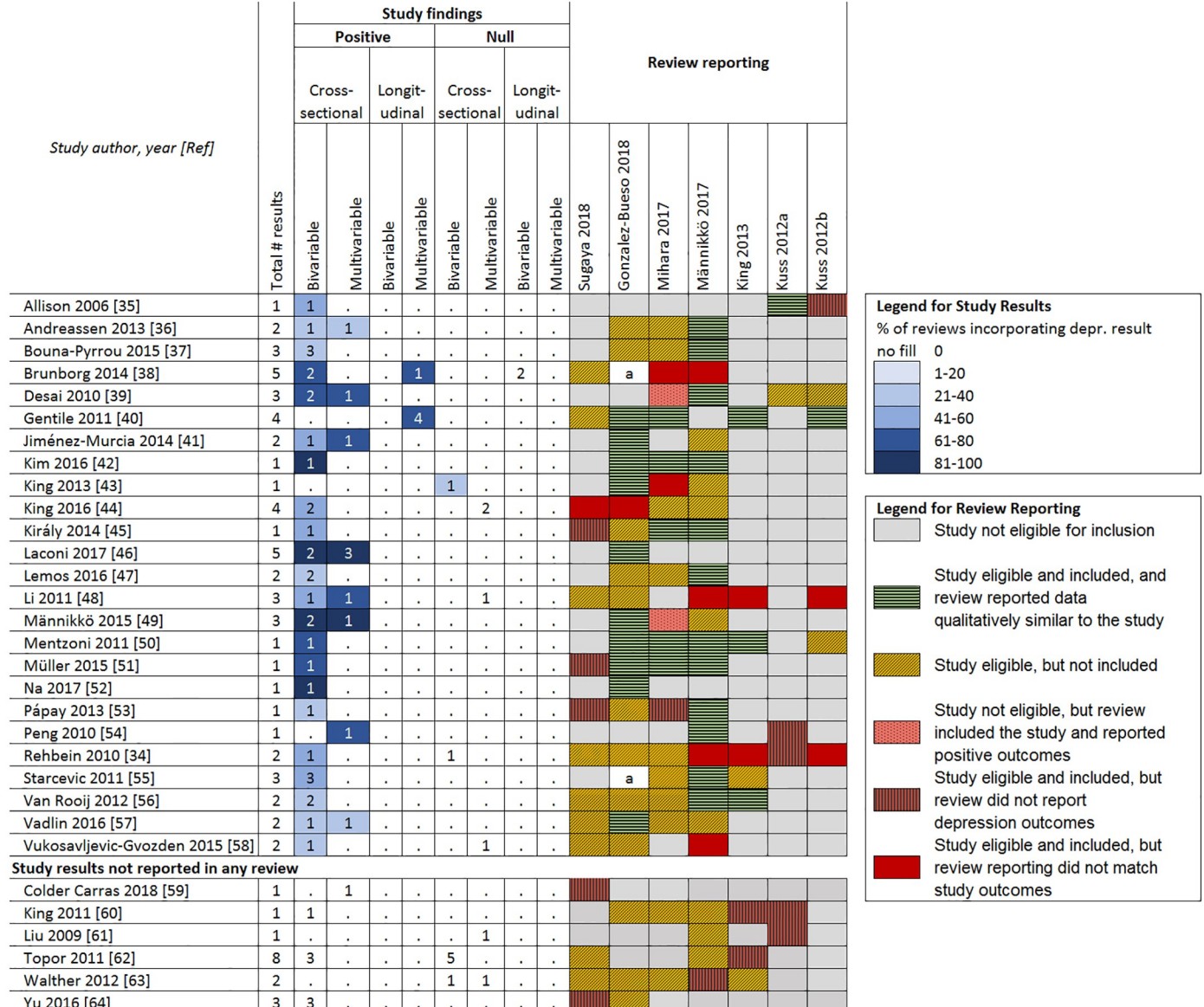

**Fig 5. Associations between problematic gaming and depression.** a = Composite reporting of outcomes in review made comparisons difficult.

## Research question 3: Reporting of associations between gaming disorder and depression or anxiety

Figs 5 and 6 report the positive and null associations for the depression and anxiety outcomes according to analysis type (bivariable/multivariable, cross-sectional, and longitudinal), their frequency of being incorporated into reviews, and how they are represented/reported in reviews (e.g., not reported, not eligible, report conflicts with outcomes). The shades of blue highlighting pertain to different percentages of reviews that incorporated a given relevant result from a given study (darker highlighting indicates higher percentages). Note that only two negative (inverse) associations were found (between gaming disorder and anxiety) and because these represent findings that were not positive and significant, they were included in

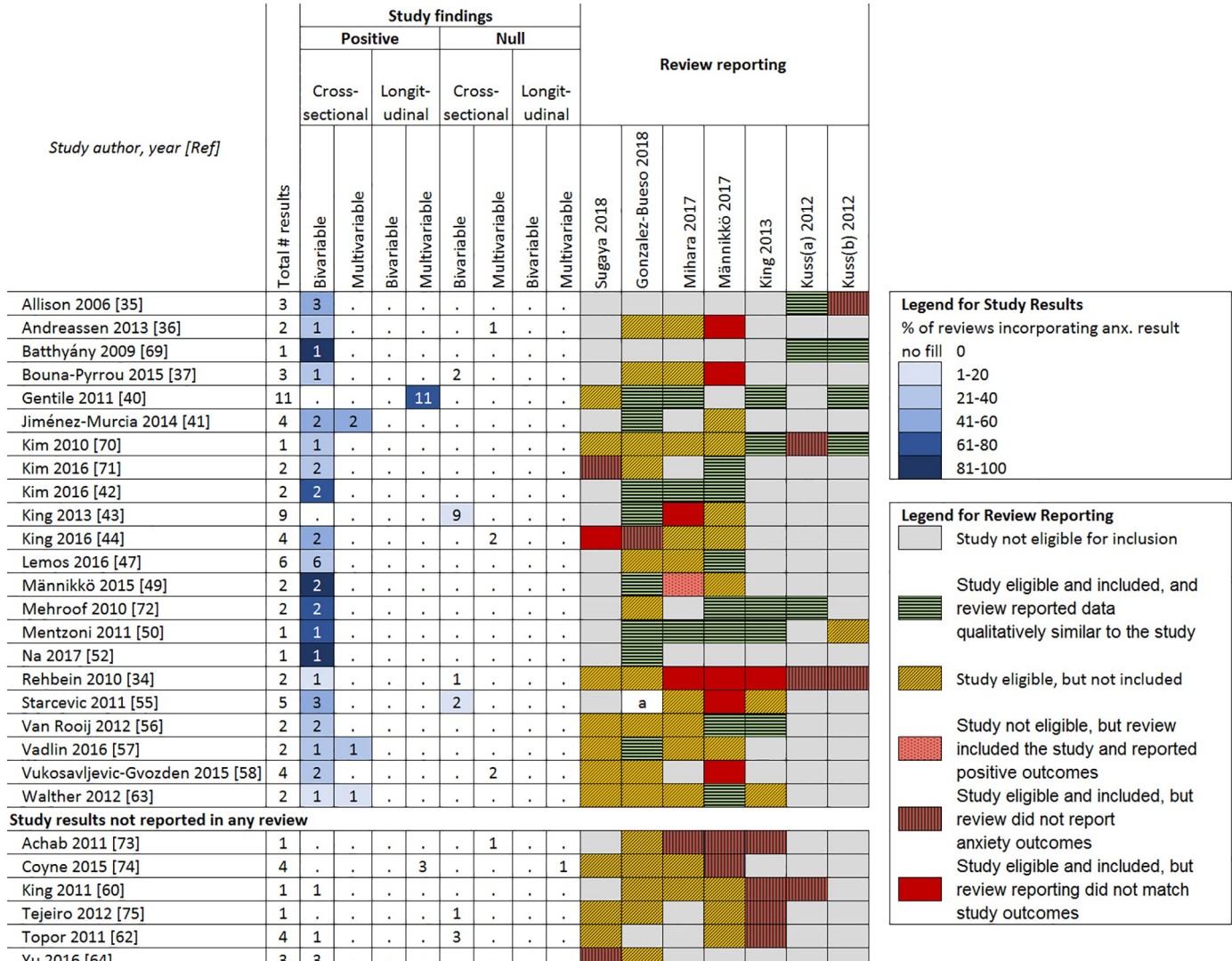

**Fig 6. Study reporting of associations between problematic gaming and anxiety.** a = Composite reporting of outcomes in review made comparisons difficult.

the count of null findings. Overall, only the review by González-Bueso and colleagues [28] reported any null results about depression or anxiety from any study.

**Associations between gaming disorder and depression.** For the depression outcome (Fig 5, including citations [34–64]), of the 31 studies reporting associations between gaming disorder and depression, results from 25 were included in at least one review. We found frequent under-incorporation of null results for the depression outcome by the reviews, as suggested by the paucity of blue cell highlights in the null columns. For example, the 2010 study by Rehbein and colleagues [34] reported two findings related to depression—a positive association between gaming disorder and suicidal thoughts in one subsample, but a null association between gaming disorder and self-reported depression in the full sample. However, the three reviews that included this study and reported results for depression all reported them as positive [30,31,33].

Ten of the 31 studies reporting associations between gaming disorder and depression reported both bivariable and multivariable analyses. In five of these 10 studies, results from

both analyses were robust and positive, while five studies reported inconsistent results. Whether consistent or inconsistent in the studies, positive results were incorporated into five of the six reviews that included depression findings from the study.

Only one study reporting an association between gaming disorder and depression examined both cross-sectional and longitudinal associations, and the results were inconsistent [38]. However, the results were incorporated into two of three reviews as showing a positive association. The final review used a composite definition when reporting associations, which made comparisons difficult [28].

Six studies reported additional cross-sectional depression results that were not incorporated into any review. Three of these studies reported null findings and in one of those cases, results were null in both bivariable and multivariable analyses. An additional 15 studies were mentioned by reviews as reporting associations between gaming disorder and depression, but using domain definitions in Figs 2 and 3, these were not found (S1 Output contains full results). All but one of the six reviews that included these studies reported these as positive associations. Some reasons for this were: studies used a measure of Internet addiction or other exposure (e.g., "excessive" gaming), studies reported a composite measure (depression/anxiety/stress) as depression, and possible mistake in citation or data extraction (e.g., reporting data for a problematic Internet use subgroup rather than problematic gaming subgroup).

In a sensitivity analysis that included studies where a broad Internet addiction scale (rather than a gaming disorder scale) was used to measure gaming disorder in a clinical population identified as having gaming disorder, one additional study [65] was found to have positive associations and was reported in the single review that included it as positive, while another three studies [66–68] had null findings which were not reported by the three reviews that included them.

**Associations between gaming disorder and anxiety.**   Of the 28 studies that reported associations between anxiety and gaming disorder, results from only 22 of these studies were incorporated into reviews (Fig 6, including citations [34–75]).

Six studies reported both bivariable and multivariable associations; half of these showed inconsistent results. Whether consistent or inconsistent, reviews incorporated only positive findings. Six studies reported results that were not incorporated into any review; four of these had inconsistent or null findings. An additional nine studies were mentioned by reviews as reporting associations between an gaming disorder and anxiety, but using domain definitions in Fig 3, these were not found. All but one of the three reviews that incorporated these studies reported these associations as positive.

In the sensitivity analysis, one additional study [65] reported inconsistent associations in bivariable and multivariable analysis and was reported as positive in the one review that contained it.

### Research question 4: Association between gaming disorder and depression or anxiety in reliable reviews

Overall, no review satisfied all the criteria we used to identify reliable reviews, so we could not address this research question.

## Discussion

This summary of systematic reviews found methodological problems in all seven systematic reviews that reported on associations between gaming disorder and depression or anxiety; no reviews could be classified as reliable based on established criteria. Although most systematic reviews studied herein defined their criteria for selecting studies and conducted a

comprehensive search, each review was rated as unreliable because of one or more of the other criteria. Because of the poor pre-specification of how outcomes would be included, it is difficult to draw conclusions from these reviews regarding associations between gaming disorder and depression or anxiety that are supported by evidence. These findings suggest that the way systematic reviews of gaming disorder have been reporting results and drawing conclusions may have introduced bias into the gaming disorder literature, possibly misleading future research, policy-making, and patient care.

Various concerns identified during this summary of systematic reviews are worthy of further discussion. We present these in the hope that the current work drives important progress in research on gaming disorder and other types of behavioral addictions in the coming years.

First, the existing reviews seldom incorporated null findings (i.e., lack of associations) or negative findings (i.e., inverse associations) from included studies even when the studies reported such findings. This is a major concern because it seems to represent selective outcome reporting at the review level. It is vital to conduct systematic reviews and meta-analyses in ways that are replicable and consistent with best practices to ensure that all evidence is reported and that relevant studies and findings are not overlooked. Selecting which outcomes of studies to include in a review without specifying the process, which has been labelled "cherry-picking" in the clinical epidemiology literature, can lead to biased conclusions at the review level [10,13]. Completely specifying all elements of outcomes (i.e., domain, specific measurements, specific metrics, methods of aggregation, and time-points of interest) or explicitly noting whether all variations of a given outcome element will be extracted is the current standard for evidence synthesis [15,26,76]. As incomplete outcome specification may lead to trillions of potential combinations of meta-analytic results [13], it is inappropriate to draw meaningful and reliable conclusions about associations between gaming disorder and the common mental health problems of depression and anxiety from the reviews summarized in this paper. Selective reporting of outcomes can be hard to detect, and further research into the impact of selective inclusion of results in reviews is needed to advance the understanding of this form of bias on evidence synthesis [77,78].

A second major concern is that reviews did not limit evidence synthesis and conclusions to studies that measured the construct of gaming disorder and at times used overly-broad definitions of depression and anxiety (e.g., combined depression, anxiety, and stress), which might have led to reports of associations between gaming disorder and depression or anxiety when none might exist.

Although more recent reviews had higher proportions of gaming-only measures, even recent reviews included studies that used Internet addiction questions to measure gaming disorder. Distinguishing between problematic behaviors is vital in ongoing research of problematic technology use and will continue to be relevant to shaping the future of health policy and government regulation of the Internet, video games, and other forms of media and technology. Ensuring that systematic and accurate measurement of gaming disorder in studies *and* accurate measurement and reporting of exposures, outcomes, and conclusions in reviews are vital to inform ongoing decision making regarding diagnosis, treatment, and public health interventions.

A third major concern is that only one review [30] reported a systematic assessment of risk of bias using multiple domains, which has long been a best practice in conducting systematic reviews [79–83]. When the risk of bias is not systematically assessed and reported, conclusions from studies included in reviews may be seen as valid and reliable when they may actually reflect biases, such as selection bias, information bias, and/or confounding [84]. When evidence of questionable methodologic quality is used to inform public health or policy decisions, such decisions may be misguided.

To our knowledge, the current analysis is the first comprehensive examination of selective outcome reporting in systematic reviews of gaming disorder, a relatively new clinical entity. Due to this selective outcome reporting, incomplete outcome specification, and lack of systematic assessment of risk of bias, we found no reviews that could be considered reliable. These findings suggest that the evidence base of systematic reviews of associations between gaming disorder and the most common mental health problems must be improved.

## Limitations

The current overview is subject to certain limitations. First, at the level of the studies we found significant inconsistencies in measurement and analysis, which were dealt with by describing counts of associations by type. While this is a somewhat reductionist approach to summarizing results, it helps paint a picture. Relatedly, no reviews defined outcomes completely. Second, we limited our analysis to systematic reviews published in English. It is possible that our findings may have been different had we included reviews in other languages. Third, we focused on the outcomes of depression or anxiety. This narrow scope made a detailed analysis possible, but findings regarding associations between gaming disorder and other outcomes (e.g., attention-deficit hyperactivity disorder) may have been different. However, due to the ubiquitous nature of selective outcome reporting, in particular, in the reviews herein, we consider this to be unlikely. Fourth, we defined the constructs of gaming disorder, depression, and anxiety very specifically; had we used broader definitions, our findings would likely be different. However, using a narrow definition was our aim. We do not attempt to draw conclusions at the study level (the 196 studies) due to the inconsistency within studies and the uncertain nature of the examined evidence. Finally, in our search of PubMed we used the PubMed publication type filters of "systematic review", "review," or "meta-analysis", while we broadened our search of PsycInfo to include these terms as text-words in all fields. For this reason, it is possible that we missed some systematic reviews that were only available in PubMed and were not indexed using these terms or did not contain these terms in the title, abstract, publication type, or keywords.

## Conclusions

To advance the field of addictive behaviors and ensure that research measures and reports constructs rigorously and with clarity, existing standards for systematic review conduct and reporting should be followed. Increasing transparency of reviews and minimizing the risk of bias requires the effort of multiple agents. Authors must prospectively register protocols (including adequately specifying outcomes); use reporting guidelines, such as those from the EQUATOR Network; and share data, analysis code, and other study materials. Journals and editors must verify authors' adherence to reporting guidelines [77]. Although public health decision-making should always proceed on the best available evidence [85], the data provided in this paper suggest that limiting technology-related diagnoses to video game play is not likely to accurately reflect the findings of years of research surrounding problematic technology use. A highly rigorous systematic review that fully specifies outcome domains is needed to clarify the potential mental health problems associated with problematic technology behaviors, including video gaming and Internet use.

## Supporting information

**S1 PRISMA Checklist. PRISMA checklist for reporting of our systematic review.**
(PDF)

**S1 Protocol. PROSPERO registration for our systematic review protocol.**
(PDF)

**S1 Search Strategy. Search strategies.**
(PDF)

**S1 Data Extraction. Data extraction at the review level, including AMSTAR 2.**
(XLSX)

**S1 Dataset. Complete analysis dataset containing extracted and derived variables.**
(DTA)

**S1 Output. Output of analysis.**
(DOCX)

## Acknowledgments

The authors are grateful to Michael M. Hughes for assistance with the graphic design and formatting of figures for publication.

## Author Contributions

**Conceptualization:** Michelle Colder Carras.

**Data curation:** Michelle Colder Carras, Jing Shi, Gregory Hard.

**Formal analysis:** Michelle Colder Carras.

**Investigation:** Michelle Colder Carras, Jing Shi, Gregory Hard, Ian J. Saldanha.

**Methodology:** Michelle Colder Carras, Jing Shi, Gregory Hard, Ian J. Saldanha.

**Project administration:** Michelle Colder Carras, Jing Shi.

**Resources:** Michelle Colder Carras, Gregory Hard, Ian J. Saldanha.

**Supervision:** Michelle Colder Carras, Jing Shi, Ian J. Saldanha.

**Validation:** Michelle Colder Carras, Jing Shi, Gregory Hard.

**Writing – original draft:** Michelle Colder Carras, Jing Shi.

**Writing – review & editing:** Michelle Colder Carras, Jing Shi, Gregory Hard, Ian J. Saldanha.

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
