## [Decision Letter · Decision Letter 0]

3 Jan 2020

PONE-D-19-32964

Evaluating the quality of evidence for gaming disorder: A systematic review of systematic reviews of associations between gaming disorder and depression or anxiety

PLOS ONE

Dear Dr Colder Carras,

Thank you for submitting your manuscript to PLOS ONE. After careful consideration, we feel that it has merit but does not fully meet PLOS ONE’s publication criteria as it currently stands. Therefore, we invite you to submit a revised version of the manuscript that addresses the points raised during the review process.

**First, I would like to thank here the two reviewers for providing very insightful comments. **The comments were all the more important for me to reach a decision, and I want you to address all of these comments in details. **Importantly, I think that some additional work is needed** : 

- Please update the searches. The topic is hot and the literature certainly expanding very fast. This is therefore important to provide up to date results. Being up to date is all the more important as your paper will attract a lot of attention ;

- Please use AMSTAR 2, as far as possible, as a secondary outcome (as suggested by reviewer 2) ; 

- Please extract information by two reviewers in an independent manner. A second reviewer must extract the information at the study level and in case of disagreement a third reviewer must arbitrate. This must be done for all the studies and not only for a sample. Vote counting is not an optimal method and therefore subject to bias during data extraction. As an editor for PLOS one I must ensure that the method is sound enough and this is an important shortcoming. Please also provide inter-rater agreement. 

- You might be interested to map the evidence base using a figure similar to Figure 2 in this reference : doi: 10.1093/ije/dyx138. Importantly, you don't have to cite this study (I'm against encouraging for self citation) but the visual display in figure 2 would be of interest in the context of your study (this is to give you an idea and it deserves surely a lot of adaptation to your specific study). To detail wether an empirical study is included in a given systematic review, a heatmap would be more readable than the current tables with numbers. In addition a plot mapping all existing empirical evidence used in the SRs across time / versus all empirical evidence available (with a line) / versus the number of empirical study in each SR would be very helpful. Indeed as it is presented now it is very difficult to understand for the reader.

- You must detail in depth the criteria used to define "an empirical" study in your methods section.

- You must provide details for selection of empirical studies in your flow charts. 

- You must search for empirical studies that were not included in the review. Indeed, all systematic reviews are incomplete in comparisons with the others as you point. But it is also important to assess wether all these SR taken together were able to find all the available empirical studies (see my comment above about the suggested figure). 

- Please make a specific section "changes to the initial protocol" to make it clear that these changes were performed during the peer review process. 

- Last please review the abstract using PRISMA statement for abstracts. Please, make sure that sufficient details are provided in the abstract and please write also a few words about the limitations of your study to avoid any over-interpretation of the findings. 

**I appreciate that the changes needed are extensive but these changes will surely help to improve the paper. **

We would appreciate receiving your revised manuscript by Feb 17 2020 11:59PM. To enhance the reproducibility of your results, we recommend that if applicable you deposit your laboratory protocols in protocols.io, where a protocol can be assigned its own identifier (DOI) such that it can be cited independently in the future. For instructions see: http://journals.plos.org/plosone/s/submission-guidelines#loc-laboratory-protocols

We look forward to receiving your revised manuscript.

Kind regards,

Florian Naudet, M.D., M.P.H., Ph.D.

Academic Editor

PLOS ONE

Journal Requirements:

Reviewers' comments:

Reviewer's Responses to Questions

**Comments to the Author**

1. Is the manuscript technically sound, and do the data support the conclusions?

Reviewer #1: Yes

Reviewer #2: Yes

2. Has the statistical analysis been performed appropriately and rigorously? 

Reviewer #1: Yes

Reviewer #2: N/A

3. Have the authors made all data underlying the findings in their manuscript fully available?

Reviewer #1: Yes

Reviewer #2: Yes

4. Is the manuscript presented in an intelligible fashion and written in standard English?

Reviewer #1: Yes

Reviewer #2: Yes

5. Review Comments to the Author

Reviewer #1: Carras et al. conducted an interesting study analyzing the quality of evidence for gaming disorder and found that no reviews were classified as reliable. This topic is of great importance and I commend the authors for taking on this task. I additionally commend the authors for following a pre-registered protocol, for reporting results according to the PRISMA reporting standards, and for making all materials available online through OSF. Overall, this is a robust study and, with minor corrections, should be considered for publication.

- The introduction and discussion is well written and appropriately introduces and discusses the topic at hand and the importance of having high-quality reviews to provide reliable evidence for healthcare and policy decision-making. I am pleased the authors have made all materials and protocols available online. This improves the transparency and reproducibility of their work.

- Line 135 says Open Science Foundation, but the correct name is Open Science Framework.

- Please provide a statement regarding IRB approval.

- While I agree the with the classification criteria chosen in this study, it would benefit from further explanation why these 6 aspects of the criteria were chosen. The authors mention Mayo-Wilson and colleagues as part of their reasoning, yet Mayo-Wilson and colleagues reference including components of the Critical Appraisal Skills Programme (CASP), the Assessment of Multiple Systematic Reviews (AMSTAR), and the Preferred Reporting Items for Systematic Reviews (PRISMA).

- The second line in the conclusion has an extra space. While the authors mention systematic reviews should follow existing standards and should be improved, this review could benefit from providing a few more concrete suggestions on how to do so.

Reviewer #2: Evaluating the quality of evidence for gaming disorder: A systematic review of systematic reviews of associations between gaming disorder and depression or anxiety

This study attempts to present a review of the quality evidence of systematic review for gaming disorder. The topic is interesting, but the rationale, the aim and the methodology of the study proposed need to be clarified.

Comments

Title

In my opinion, the proposed title is inconsistent with the main aim of the study. After reading all the manuscript, I am wondering if your main research question is relative to selective reporting of outcomes, or if it rather of focused on the quality of evidence in systematic reviews.

Abstract

Currently, there are opportunities to improve the understanding of the abstract. The terminology used is not clear and the aim of the study is not clearly defined either.

Introduction

In my opinion, the rationale for this study could be improved. This section should be edited explaining the background to the study, an accurate summary of the existing literature and why this study is necessary.

Some specific comments:

1. I think the second part of the introduction “Synthesizing evidence from systematic reviews” needs to be improved and reworked. Moreover, I do not totally agree with your statement (line 85-90), they seem to suggest that the reliability of a systematic review is only due to the definition of an outcome. Please, revise your definition.

2. In my opinion, the figure 1 should not be presented in the introduction part but in the methodological part as a rational to characterize an outcome.

3. Regarding your sentence in line 96-98, I think that to claim “Reviews that actively minimize bias and are reported transparently can be considered high-quality” could be (too) over-simplifying the methodological quality of a well conduct systematic review.

4. In my opinion, the rationale for this study is not clear (and accurate). This section should explain the background to the study, its aims, an accurate summary of the existing literature and why this study is necessary or its contribution to the field. Specifically, authors should describe the rationale for the analysis in the context of what is already known.

Methods

1. Page 9-10: Why did the authors choose the scale proposed by Mayo-Wilson and colleagues? Some others tools could be more suited (for example the tool AMSTAR2). I am very surprised by the way you classified the review as reliable. Please provide a rationale and more explanation about the use of the tool.

2. Page 11, search strategy: Usually, it is recommended not to use the publication type as limitation to be the most exhaustive. Moreover, to be “systematic”, the methodology of a systematic review requires consulting also grey literature (for example by checking the reference list of the retrieved article or by searching the congress abstract). Did the authors consult other sources than scientific database? It could be a limitation of your paper.

3. Page 11, line 215-217: What was the reliability of the coding? Did the authors calculated a Kappa coefficient to justify the validation of a random sample of 10%?

4. Page 12, line 242: See my comment above (Page9-10, #1).

5. Page 13, line 252-258: Regarding your “vote counting approach”, I am not sure to really understand the aim of this approach. Could you provide more information about this method and the rationale to use this approach?

Results

1. I am very surprised that the authors found only 6 studies. Have you been the most exhaustive? See my comment above (Introduction #4 and method #2).

2. The legend for table 1 is missing.

3. Because the primary outcome of this review is to assess the review quality, the result for this part should be more detailed. In my opinion, the result presented do not really represent the review quality of a systematic review. Please, revise.

4. In table 2, It could be relevant to add “empirical study”. The difference between review and “empirical study” is not clear in the result part.

Discussion and conclusions

1. I am wondering about your rationale and your methodology to assess the selective outcome reporting in the review. In my opinion, the presentation of an outcome or not, depends of the aim of the review. All the outcome presented in the empirical study are not always necessary presented in the review. Did the authors check the presence of a research protocol of the 6 reviews? It could be helpful to assess the selective outcome reporting (for example: to identify the difference between the outcome specify in the research protocol and the outcome presented in the review paper).

Thank you for the opportunity to review this paper. I look forward to receiving feedback about the

peer-review process of this manuscript.

6. PLOS authors have the option to publish the peer review history of their article (what does this mean?). If published, this will include your full peer review and any attached files.

Reviewer #1: Yes: Austin L. Johnson

Reviewer #2: No

---

## [Author Response · Author response to Decision Letter 0]

27 Jul 2020

Please also see the submitted file Response to reviewers. Text is pasted below.

- Please update the searches. The topic is hot and the literature certainly expanding very fast. This is therefore important to provide up to date results. Being up to date is all the more important as your paper will attract a lot of attention ;

Response: We have updated our searches to June 24, 2020.

- Please use AMSTAR 2, as far as possible, as a secondary outcome (as suggested by reviewer 2) ; 

Response: AMSTAR 2 has now been applied and is available in S3 Data extraction.

- Please extract information by two reviewers in an independent manner. A second reviewer must extract the information at the study level and in case of disagreement a third reviewer must arbitrate. This must be done for all the studies and not only for a sample. Vote counting is not an optimal method and therefore subject to bias during data extraction. As an editor for PLOS one I must ensure that the method is sound enough and this is an important shortcoming. Please also provide inter-rater agreement. 

Response: Unfortunately, we do not have the resources do independent verification of extracted data for all primary studies. We have addressed this by revising the description of our study to that of a summary of systematic reviews. We acknowledge this as a limitation in the Discussion section. That being said, we extracted data at the review level in duplicate, but do did not calculate inter-rater agreement. 

- You might be interested to map the evidence base using a figure similar to Figure 2 in this reference : doi: 10.1093/ije/dyx138. Importantly, you don't have to cite this study (I'm against encouraging for self citation) but the visual display in figure 2 would be of interest in the context of your study (this is to give you an idea and it deserves surely a lot of adaptation to your specific study). To detail wether an empirical study is included in a given systematic review, a heatmap would be more readable than the current tables with numbers. In addition a plot mapping all existing empirical evidence used in the SRs across time / versus all empirical evidence available (with a line) / versus the number of empirical study in each SR would be very helpful. Indeed as it is presented now it is very difficult to understand for the reader.

Response: This is a very helpful graphic. We have incorporated a heatmap-type presentation of these findings (see Tables 2 and 3) and descriptions of them in the Results section. We believe that this and other improvements to the manuscript greatly improve its quality and reporting.

- You must detail in depth the criteria used to define "an empirical" study in your methods section.

Response: We did not actually choose which empirical studies to include; we only used those included in the reviews. This is clarified in the Methods section.

- You must provide details for selection of empirical studies in your flow charts. 

Response: See response to previous comment.

- You must search for empirical studies that were not included in the review. Indeed, all systematic reviews are incomplete in comparisons with the others as you point. But it is also important to assess wether all these SR taken together were able to find all the available empirical studies (see my comment above about the suggested figure). 

Response: Our goal was to assess the reliability of reviews based on defined criteria. While we feel it is important to ensure reviews capture relevant studies, to do a separate systematic review for all empirical studies would be outside the scope of the present study.

- Please make a specific section "changes to the initial protocol" to make it clear that these changes were performed during the peer review process. 

As we did not make the changes suggested above, we did not include a section "changes to the initial protocol".

- Last please review the abstract using PRISMA statement for abstracts. Please, make sure that sufficient details are provided in the abstract and please write also a few words about the limitations of your study to avoid any over-interpretation of the findings. 

Response: We have made the recommended changes to the abstract. 

I appreciate that the changes needed are extensive but these changes will surely help to improve the paper. 

Response: We are grateful for your careful review and suggestions.

5. Review Comments to the Author

Reviewer #1: Carras et al. conducted an interesting study analyzing the quality of evidence for gaming disorder and found that no reviews were classified as reliable. This topic is of great importance and I commend the authors for taking on this task. I additionally commend the authors for following a pre-registered protocol, for reporting results according to the PRISMA reporting standards, and for making all materials available online through OSF. Overall, this is a robust study and, with minor corrections, should be considered for publication.

Response: Thank you.

- The introduction and discussion is well written and appropriately introduces and discusses the topic at hand and the importance of having high-quality reviews to provide reliable evidence for healthcare and policy decision-making. I am pleased the authors have made all materials and protocols available online. This improves the transparency and reproducibility of their work.

Response: Thank you.

- Line 135 says Open Science Foundation, but the correct name is Open Science Framework.

Response: We have made that correction.

- Please provide a statement regarding IRB approval.

Response: We have added that on page 14 at the end of the Methods section.

- While I agree the with the classification criteria chosen in this study, it would benefit from further explanation why these 6 aspects of the criteria were chosen. The authors mention Mayo-Wilson and colleagues as part of their reasoning, yet Mayo-Wilson and colleagues reference including components of the Critical Appraisal Skills Programme (CASP), the Assessment of Multiple Systematic Reviews (AMSTAR), and the Preferred Reporting Items for Systematic Reviews (PRISMA).

Response: Our goal was to apply an existing approach to determining reliability of reviews to an entirely new topic area. We described this in more detail on pages 9-10 and included the full AMSTAR 2 assessment in S3 Data extraction.

- The second line in the conclusion has an extra space. 

Response: We made sure there were no extra spaces.

While the authors mention systematic reviews should follow existing standards and should be improved, this review could benefit from providing a few more concrete suggestions on how to do so.

Response: We have made additional suggestions as to how systematic review authors could improve their reviews on pages 29-30.

Reviewer #2: Evaluating the quality of evidence for gaming disorder: A systematic review of systematic reviews of associations between gaming disorder and depression or anxiety

This study attempts to present a review of the quality evidence of systematic review for gaming disorder. The topic is interesting, but the rationale, the aim and the methodology of the study proposed need to be clarified.

Response: We have revised the manuscript extensively to conform with the reviewer's suggestions.

Comments

Title

In my opinion, the proposed title is inconsistent with the main aim of the study. After reading all the manuscript, I am wondering if your main research question is relative to selective reporting of outcomes, or if it rather of focused on the quality of evidence in systematic reviews.

Response: The title now reflects that we are summarizing systematic reviews. However, our research questions remain unchanged. Our first question addresses quality, while subsequent questions address aspects of quality that may be lacking, such as conflation of domains of the exposure (gaming disorder/IGD/etc) and selective outcome reporting. 

Abstract

Currently, there are opportunities to improve the understanding of the abstract. The terminology used is not clear and the aim of the study is not clearly defined either.

Response: The abstract has been extensively revised and reported in accordance with the PRISMA for Abstracts reporting guideline. 

Introduction

In my opinion, the rationale for this study could be improved. This section should be edited explaining the background to the study, an accurate summary of the existing literature and why this study is necessary.

Response: We have revised the Introduction extensively to conform with the reviewer's suggestions.

Some specific comments:

1. I think the second part of the introduction “Synthesizing evidence from systematic reviews” needs to be improved and reworked. Moreover, I do not totally agree with your statement (line 85-90), they seem to suggest that the reliability of a systematic review is only due to the definition of an outcome. Please, revise your definition.

Response: The revised Introduction section now provides a clear and comprehensive description of various relevant potential sources of bias in systematic reviews and the revised Methods section now clarifies how our study assesses reliability of systematic reviews.

2. In my opinion, the figure 1 should not be presented in the introduction part but in the methodological part as a rational to characterize an outcome.

Response: Figure 1 is an important piece of background information that many who study gaming disorder but are not systematic review specialists may not be familiar with. It describes how outcomes are specified in general. We have retained this in the Introduction but have added a second figure to the Methods section that more clearly describes how we defined the broad domains of gaming disorder, depression and anxiety in our data extraction.

3. Regarding your sentence in line 96-98, I think that to claim “Reviews that actively minimize bias and are reported transparently can be considered high-quality” could be (too) over-simplifying the methodological quality of a well conduct systematic review.

Response: We have removed that statement. 

4. In my opinion, the rationale for this study is not clear (and accurate). This section should explain the background to the study, its aims, an accurate summary of the existing literature and why this study is necessary or its contribution to the field. Specifically, authors should describe the rationale for the analysis in the context of what is already known.

Response: We have extensively added text to the Introduction to address these issues. As we explain in the Introduction on page 7 and the Discussion on page 28, our study is the first comprehensive examination of selective outcome reporting in systematic reviews of gaming disorder, a new clinical entity. There have been no prior studies focusing on any type of bias assessment in systematic reviews of gaming disorder. 

Methods

1. Page 9-10: Why did the authors choose the scale proposed by Mayo-Wilson and colleagues? Some others tools could be more suited (for example the tool AMSTAR2). I am very surprised by the way you classified the review as reliable. Please provide a rationale and more explanation about the use of the tool.

Response: As we mention in our response to Reviewer 1, our goal with this was to apply an existing approach to determining reliability of reviews to an entirely new topic area. We adopted the definition of reliable systematic reviews that has been used quite widely. We described this in more detail on pages 9-10 and included the full AMSTAR 2 assessment in S3 Data extraction.

2. Page 11, search strategy: Usually, it is recommended not to use the publication type as limitation to be the most exhaustive. 

Response: We recognize that the use of review publication type filters is a limitation; we mention it as a limitation in the Discussion section on page 29. 

Moreover, to be “systematic”, the methodology of a systematic review requires consulting also grey literature (for example by checking the reference list of the retrieved article or by searching the congress abstract). Did the authors consult other sources than scientific database? It could be a limitation of your paper.

Response: We have improved our methods and now include a grey literature search, as described on page 9. 

3. Page 11, line 215-217: What was the reliability of the coding? Did the authors calculated a Kappa coefficient to justify the validation of a random sample of 10%?

Response: We did not estimate Kappa coefficients, but did a few rounds of pilot-testing. Estimation of Kappa coefficients is not widely recommended for screening and data extraction.

4. Page 12, line 242: See my comment above (Page9-10, #1). 

Response: Our revised manuscript clarifies that our focus on quality assessment (pages 9-10) addresses risk of bias.

5. Page 13, line 252-258: Regarding your “vote counting approach”, I am not sure to really understand the aim of this approach. Could you provide more information about this method and the rationale to use this approach?

Response: Instead of vote counting, we present tallies of positive, negative, and null results for transparency. These can be compared directly to results found in the individual studies of the reviews we summarized.

Results

1. I am very surprised that the authors found only 6 studies. Have you been the most exhaustive? See my comment above (Introduction #4 and method #2).

Response: Yes, our search was comprehensive, including a search of multiple electronic databases and a grey literature search. We have now updated our search and found one more relevant systematic review. We also present a list of excluded studies with reasons for exclusion on the OSF webpage for the project. 

2. The legend for table 1 is missing.

Response: All tables now have legends.

3. Because the primary outcome of this review is to assess the review quality, the result for this part should be more detailed. In my opinion, the result presented do not really represent the review quality of a systematic review. Please, revise.

Response: This section has been extensively revised. We have clarified the assessment of review quality (reliability).

4. In table 2, It could be relevant to add “empirical study”. The difference between review and “empirical study” is not clear in the result part.

Response: We now refrain from using the ‘empirical’ when denoting a study or a review because it is an ambiguous term. Early in the Methods section, we define the eligible systematic reviews and the primary studies they included as ‘reviews’ and ‘studies’, respectively.

Discussion and conclusions

1. I am wondering about your rationale and your methodology to assess the selective outcome reporting in the review. In my opinion, the presentation of an outcome or not, depends of the aim of the review. All the outcome presented in the empirical study are not always necessary presented in the review. Did the authors check the presence of a research protocol of the 6 reviews? It could be helpful to assess the selective outcome reporting (for example: to identify the difference between the outcome specify in the research protocol and the outcome presented in the review paper).

Response: In the Introduction, we have bolstered the argument for the need for assessing selective outcome reporting. One problem that we were trying to address is the lack of attention to standards for systematic reviews in this area. Of the now seven systematic reviews included in the manuscript, only one included a protocol, and indeed this was the only review to specify which outcomes would be reported and to report any non-positive associations. We have noted the lack of pre-specifications in the AMSTAR 2 reporting now found in S3 Review data extraction.

Thank you for the opportunity to review this paper. I look forward to receiving feedback about the peer-review process of this manuscript.

 Response: Thank you for your helpful comments.

---

## [Decision Letter · Decision Letter 1]

2 Sep 2020

PONE-D-19-32964R1

Evaluating the quality of evidence for gaming disorder: A summary of systematic reviews of associations between gaming disorder and depression or anxiety

PLOS ONE

Dear Dr. Colder Carras,

Thank you for submitting your manuscript to PLOS ONE. After careful consideration, we feel that it has merit but does not fully meet PLOS ONE’s publication criteria as it currently stands. Therefore, we invite you to submit a revised version of the manuscript that addresses the points raised during the review process.

I would like to thank the 2 reviewers. Thank you for the changes that were made. One of the two suggest that there are still important points to address and I agree. In addition, Table 2 needs to be attached as a Figure. Idem for table 3. Else, I suspect that the editorial staff will not be able to used colors in a Table (and with such colors, it is now rather a figure than a table).

We look forward to receiving your revised manuscript.

Kind regards,

Florian Naudet, M.D., M.P.H., Ph.D.

Academic Editor

PLOS ONE

Reviewers' comments:

Reviewer's Responses to Questions

**Comments to the Author**

1. If the authors have adequately addressed your comments raised in a previous round of review and you feel that this manuscript is now acceptable for publication, you may indicate that here to bypass the “Comments to the Author” section, enter your conflict of interest statement in the “Confidential to Editor” section, and submit your "Accept" recommendation.

Reviewer #1: All comments have been addressed

Reviewer #2: All comments have been addressed

2. Is the manuscript technically sound, and do the data support the conclusions?

Reviewer #1: Yes

Reviewer #2: Yes

3. Has the statistical analysis been performed appropriately and rigorously? 

Reviewer #1: Yes

Reviewer #2: Yes

4. Have the authors made all data underlying the findings in their manuscript fully available?

Reviewer #1: Yes

Reviewer #2: Yes

5. Is the manuscript presented in an intelligible fashion and written in standard English?

Reviewer #1: Yes

Reviewer #2: Yes

6. Review Comments to the Author

Reviewer #1: (No Response)

Reviewer #2: This study attempts to present a review of the quality evidence of systematic review for gaming disorder. I also reviewed a previous version of this manuscript. The authors did a good job in revising the manuscript, but I still have some minus remaining comments.

Comments

1. There is a syntax error in the sentence on line 167 page 9.

2. The authors have chosen to use your own definition using various sources to determine a reliable systematic review. Therefore, I think it would be useful for a better understanding of all, to better define your 6 criteria. For example: how did you judge the criterion 'comprehensive literature search'? Did you give a yes when the authors had consulted at least one database and had not limited their searches in English or other criteria were taken into account? The remark is valid for all items. How did you assign yes and no to the different items for each systematic review?

3. Thank you for appending the search strategy used in Medline. Could you add the one used in Psycinfo to be completely transparent?

4. Finally, the authors used the AMSTAR2 tool to assess the methodological quality of systematic reviews as outlined in the method. However, the results are not presented even briefly in the results section, let alone discussed in the discussion. A commentary on these results would be very interesting and could allow a comparison of the results with the existing literature on this topic.

7. PLOS authors have the option to publish the peer review history of their article (what does this mean?). If published, this will include your full peer review and any attached files.

Reviewer #1: No

Reviewer #2: No

---

## [Author Response · Author response to Decision Letter 1]

14 Sep 2020

We respond here to revision requests and suggestions, but this information can also be found in the file "Response to reviewers". Thank you again for your assistance in improving our manuscript for publication. 

6. Review Comments to the Author

Reviewer #1: (No Response)

Reviewer #2: This study attempts to present a review of the quality evidence of systematic review for gaming disorder. I also reviewed a previous version of this manuscript. The authors did a good job in revising the manuscript, but I still have some minus remaining comments.

Comments

1. There is a syntax error in the sentence on line 167 page 9.

Response: Thank you for bringing this to our attention. We have fixed this error.

2. The authors have chosen to use your own definition using various sources to determine a reliable systematic review. Therefore, I think it would be useful for a better understanding of all, to better define your 6 criteria. For example: how did you judge the criterion 'comprehensive literature search'? Did you give a yes when the authors had consulted at least one database and had not limited their searches in English or other criteria were taken into account? The remark is valid for all items. How did you assign yes and no to the different items for each systematic review?

Response: We have clarified the definition of reliability. As we stated in the opening sentence of the section “Assessment of reliability of reviews”, this definition was not created de novo, but was adapted from one developed and published by Cochrane Eyes and Vision. Further, we have made changes throughout the manuscript to avoid confusion between “reliability” and “quality”. 

3. Thank you for appending the search strategy used in Medline. Could you add the one used in Psycinfo to be completely transparent?

Response: We have now also appended the search strategy that we used for searching PsycInfo (in S2 (Search Strategies)).

4. Finally, the authors used the AMSTAR2 tool to assess the methodological quality of systematic reviews as outlined in the method. However, the results are not presented even briefly in the results section, let alone discussed in the discussion. A commentary on these results would be very interesting and could allow a comparison of the results with the existing literature on this topic.

Response: As mentioned in our response to your comment #2 above, we have clarified the distinction between ‘reliability’ (assessed using a few items from AMSTAR and other tools) and ‘quality’ (assessed using all AMSTAR items). The focus of this paper is on reliability of reviews, so we describe the AMSTAR items relevant to reliability in the main paper, and we include the results of the full AMSTAR assessment in the appendix table (S3).

---

## [Decision Letter · Decision Letter 2]

18 Sep 2020

Evaluating the quality of evidence for gaming disorder: A summary of systematic reviews of associations between gaming disorder and depression or anxiety

PONE-D-19-32964R2

Dear Dr. Colder Carras,

We’re pleased to inform you that your manuscript has been judged scientifically suitable for publication and will be formally accepted for publication once it meets all outstanding technical requirements.

Kind regards,

Florian Naudet, M.D., M.P.H., Ph.D.

Academic Editor

PLOS ONE

Additional Editor Comments (optional):

Reviewers' comments:

Reviewer's Responses to Questions

**Comments to the Author**

1. If the authors have adequately addressed your comments raised in a previous round of review and you feel that this manuscript is now acceptable for publication, you may indicate that here to bypass the “Comments to the Author” section, enter your conflict of interest statement in the “Confidential to Editor” section, and submit your "Accept" recommendation.

Reviewer #2: All comments have been addressed

2. Is the manuscript technically sound, and do the data support the conclusions?

Reviewer #2: Yes

3. Has the statistical analysis been performed appropriately and rigorously? 

Reviewer #2: Yes

4. Have the authors made all data underlying the findings in their manuscript fully available?

Reviewer #2: Yes

5. Is the manuscript presented in an intelligible fashion and written in standard English?

Reviewer #2: Yes

6. Review Comments to the Author

Reviewer #2: I have no further comments. Good work.

7. PLOS authors have the option to publish the peer review history of their article (what does this mean?). If published, this will include your full peer review and any attached files.

Reviewer #2: No

---

## [Editor Report · Acceptance letter]

24 Sep 2020

PONE-D-19-32964R2 

Evaluating the quality of evidence for gaming disorder: A summary of systematic reviews of associations between gaming disorder and depression or anxiety 

Dear Dr. Colder Carras:

I'm pleased to inform you that your manuscript has been deemed suitable for publication in PLOS ONE. Congratulations! Your manuscript is now with our production department. 

Kind regards, 

on behalf of

Pr. Florian Naudet 

Academic Editor

PLOS ONE